# Pathological Role of Oxidative Stress in Aflatoxin-Induced Toxicity in Different Experimental Models and Protective Effect of Phytochemicals: A Review

**DOI:** 10.3390/molecules28145369

**Published:** 2023-07-13

**Authors:** Martha Cebile Jobe, Doctor M. N. Mthiyane, Phiwayinkosi V. Dludla, Sithandiwe E. Mazibuko-Mbeje, Damian C. Onwudiwe, Mulunda Mwanza

**Affiliations:** 1Department of Animal Science, Mahikeng Campus, North-West University, Mmabatho 2735, South Africa; 36918067@student.g.nwu.ac.za (M.C.J.);; 2Food Security and Safety Focus Area, Mahikeng Campus, North-West University, Mmabatho 2735, South Africa; 3Cochrane South Africa, South African Medical Research Council, Tygerberg 7505, South Africa; 4Department of Biochemistry and Microbiology, University of Zululand, KwaDlangezwa 3886, South Africa; 5Department of Biochemistry, Mahikeng Campus, North-West University, Mmabatho 2735, South Africa; 6Department of Chemistry, Mahikeng Campus, North-West University, Mmabatho 2735, South Africa; 7Department of Animal Health, Mahikeng Campus, North-West University, Mmabatho 2735, South Africa

**Keywords:** aflatoxins, oxidative stress, phytochemicals, antioxidants, broilers

## Abstract

Aflatoxin B1 is a secondary metabolite with a potentially devastating effect in causing liver damage in broiler chickens, and this is mainly facilitated through the generation of oxidative stress and malonaldehyde build-up. In the past few years, significant progress has been made in controlling the invasion of aflatoxins. Phytochemicals are some of the commonly used molecules endowed with potential therapeutic effects to ameliorate aflatoxin, by inhibiting the production of reactive oxygen species and enhancing intracellular antioxidant enzymes. Experimental models involving cell cultures and broiler chickens exposed to aflatoxin or contaminated diet have been used to investigate the ameliorative effects of phytochemicals against aflatoxin toxicity. Electronic databases such as PubMed, Science Direct, and Google Scholar were used to identify relevant data sources. The retrieved information reported on the link between aflatoxin B1-included cytotoxicity and the ameliorative potential/role of phytochemicals in chickens. Importantly, retrieved data showed that phytochemicals may potentially protect against aflatoxin B1-induced cytotoxicity by ameliorating oxidative stress and enhancing intracellular antioxidants. Preclinical data indicate that activation of nuclear factor erythroid 2-related factor 2 (Nrf2), together with its downstream antioxidant genes, may be a potential therapeutic mechanism by which phytochemicals neutralize oxidative stress. This highlights the need for more research to determine whether phytochemicals can be considered a useful therapeutic intervention in controlling mycotoxins to improve broiler health and productivity.

## 1. Introduction

Mycotoxins are classified as a wide range of harmful compounds, with aflatoxins, fumonisin, trichothecenes, ochratoxins, and zearalenone, with aflatoxin being the most studied. The harmful effects of mycotoxins have led regulatory entities, such as the Food Agriculture Organization and the World Health Organization, to establish regulatory measurements to monitor and control the levels of mycotoxins in foods and feed [1,2]. The term “mycotoxicosis” emerged for the first time in 1962 in a study of animal sickness, following the deaths of approximately 100,000 young turkeys in the United Kingdom in 1960 [3,4]. The toxin that caused poultry deaths was linked to *Aspergillus flavus* that was isolated from the feed; hence, the discovery of aflatoxins marked the beginning of contemporary mycotoxin study. Over the years, research and field investigations have revealed that mycotoxins, which are secondary metabolites produced by some fungal species, are responsible for both food spoilage and the development of many diseases [5,6]. Aflatoxin B1 (AFB1) is well-known among the different aflatoxin types because of its widespread occurrence, high toxicity, and economic implications across the world [7]. According to the guidelines of the International Agency for Research on Cancer (IARC), AFB1 is classified as Group 1 [8], with widely reported hepatotoxic, carcinogenic, teratogenic, and immunosuppressive effects in mammals and poultry at large [9,10,11,12].

This has led to a significant interest in understanding the health implications of aflatoxins, with the PubMed search showing over 1493 records on this carcinogen, suggesting a considerable growth of publications over the years [13,14,15]. At present, oxidative stress appears to be one of the prime mechanisms of AFB1-induced toxicity in various disease settings. Briefly, exposure to AFB1 can induce an abnormally increased formation of free radicals, hence generating lipid peroxidation through their reaction with lipid products in the body: a consequence that leads to massive cellular damage which ultimately causes death to both animals and humans [16,17]]. Even more specifically for animals, AFB1 can induce the overproduction of toxic reactive oxygen species (ROS), which can also contribute to the formation of oxidative stress [18,19]. Apparently, this process can activate the cytochrome P450 (CYP450) enzyme system and produce AFB1-8, 9-epoxide, a toxic metabolite of AFB1 [20]. As a result, the formation of DNA adducts can therefore lead to genetic changes and cause the transformation of liver hepatocytes, which is the major detoxification hub within the body [21].

Several physical, chemical, and biological methodologies have been explored to eliminate aflatoxins, as the control of mycotoxins is a critical aspect of nutrition research [22,23,24]. Consistently, several compounds have been used to reduce aflatoxin toxicity in poultry feed. Such compounds include zeolites [25], sodium bentonite [26], and mannose oligosaccharides [27]. On another hand, experimental results are recommending the use of phytochemicals because they can ameliorate aflatoxin toxicity in experimental models [28]. These compounds can act as modulators of gene expression and/or enzymatic activities of biotransformation enzymes involved in aflatoxin activation and detoxification [29,30,31,32]. Indeed, some plant products contain abundant phytochemicals that can be effective against aflatoxin toxicity in broilers [33]. This is especially relevant since phytochemicals are known to contain abundant antioxidant properties that are essential in protecting against oxidative stress-induced cellular damage [34,35]. For example, increasing interest has been placed on unravelling the therapeutic effects of phytochemicals like berberine [36], carvacrol, curcumin [37,38], proanthocyanidins [39,40], quercetin [41,42], resveratrol [43], and silymarin [44,45,46] in protecting against oxidative stress, leading to alleviation of cell damage in many experimental models [47,48]. In fact, such information has not been comprehensively reviewed in relation to aflatoxin-induced toxicity.

Thus, the current review brings a new perspective on the role of oxidative stress in aflatoxin-induced cellular damage in different experimental models. Special focus is placed on discussing how different or prominently discussed phytochemicals and dietary compounds can be used to protect against aflatoxin-induced toxicity, especially in broiler chickens. Therefore, a brief overview of the role of oxidative stress in aflatoxin toxicity is covered, before a methodology is given describing how the literature was searched and selected based on the impact of phytochemicals in protecting against aflatoxin-induced toxicity.

## 2. An Overview of Oxidative Stress and Its Role in Aflatoxin Toxicity

Oxidative stress defines an imbalance between the production of free radicals, including ROS, and the counteractive protective effects of intracellular antioxidant systems. Thus far, many studies have been published informing on the important role oxidative stress plays in the development and progression of various diseases [49,50,51]. While free radicals are essential for an efficient physiological process, their overproduction has been linked with the pathological state, through their attack on essential biomolecules including proteins, carbohydrates, lipids, and nucleic acids [52,53]. This process is known to have a devastating outcome in vital biochemical processes and cell signaling mechanisms that eventually render cellular damage. For example, lipid peroxidation, due to oxidants such as free radicals pouncing on lipids containing carbon–carbon double bonds [54], has long been linked to liver damage in broiler chickens poisoned with the antibiotic monensin, which is derived from the bacterium *Streptomyces cinnamonensis* [55]. Indeed, many internal and external factors are implicated in the production of free radicals that accelerate oxidative damage. In broiler chickens, for example, aflatoxin or its metabolites are directly induced through enhanced production of free radicals and lipid peroxides, resulting in cell damage [56,57]. An aspect of particular interest is hepatotoxicity, since liver CYP450 enzyme systems can metabolize aflatoxins, leading to chain activation of other free radical species like superoxide radicals and hydrogen peroxide [37,58]. In brief, it is well established that aflatoxin can be metabolized in the liver to its reactive form, AFB1-8,9-epoxide, which produces adducts upon reacting with both DNA and protein [59]. Subsequently, this process is instrumental in exacerbating apoptosis and facilitating cellular damage.

Consumption of AFB1-contaminated feed has been linked to toxicity in broilers [60], immunosuppression [61], and increased disease susceptibility [58,62]. The harmful effect of aflatoxins was examined using in vivo and in vitro experimental models see Table 1, reporting on the development of oxidative stress that is consistent with the exacerbation of hepatoxicity [63,64]. The weakening of antioxidant defences in murine livers, human lymphocytes, and bovine peripheral blood mononuclear cells, due to AFB1-induced generation of free radicals, has also been reported [65,66,67]. Oxidative stress in broilers has also been linked to the accumulation of AFB1 absorbed from the gastrointestinal tract causing toxicity in the liver, the main detoxification organ [10]. Following liver damage, the metabolism of proteins, carbohydrates, and lipids is hampered. An increase in free radical production due to AFB1 leads to lipid peroxidation in broiler chickens [68,69], with concomitant depletion of tissue/cellular sulphydryl forms of thiols. Glutathione (GSH) is the most important non-enzymatic antioxidant and should be considered an early biological marker of oxidative stress [70]. The conjugation of active AFB1 with GSH, which is mediated by glutathione S-transferases (GST), is a crucial process in the detoxification and excretion of this toxin [71]. Concerning that, poultry species have phase-I liver enzymes for activating AFB1 but weak phase-II GST enzyme productivity for detoxifying AFB1 and its toxic metabolites [72]. Importantly, AFB1 mainly causes an increase in malondialdehyde (MDA) concentrations, which may lead to a decrease in enzymatic antioxidants such as catalase (CAT), glutathione peroxidase (GPx), and superoxide dismutase (SOD) [37]. The current report discusses published information on the role of oxidative stress in broiler chickens, especially how it is influenced by the aflatoxin-contaminated diet or exposure to it. Moreover, it remains essential to understand whether the activation of nuclear factor erythroid 2-related factor 2 (Nrf2), the main intracellular antioxidant response mechanism, can ameliorate or protect against oxidative stress in relation to aflatoxins [33,62,73]. Importantly, Nrf2 can be activated by some phytochemicals and this mechanism is known to control the expression of a group of antioxidants and detoxification genes that give a protective role against oxidative stress and has been established against multiple diseases such as cancer, pulmonary disease, and inflammation [74,75].

## 3. Methodology for Literature Search and Study Inclusion

Briefly, a comprehensive search was undertaken on some of the major electronic databases such as PubMed, Science Direct, and Google Scholar to find relevant experimental studies. This included the use of medical subject heading (MeSH) words such as “aflatoxins”, “oxidative stress”, “phytochemicals”, and “broilers”, including relevant synonyms, which were used to optimize the search. There was no language restriction. This review included both in vitro and in vivo studies, while reviews were screened for primary data, reporting on the link between aflatoxins and oxidative stress in various experimental models. In fact, the focus was on studies that reported on broiler chickens. In a nutshell, the goal of this review was to have a better understanding of the effects of aflatoxins and phytochemicals in ethnoveterinary practices, human studies, and case reports. Therefore, the following data items were extracted, including the experimental model, study design, animal model, doses, principal findings, and author information (name and year of the publication). The below data give an overview of the literature reporting on the impact of phytochemicals on aflatoxin toxicity. This includes both in vitro and in vivo studies revealing ameliorating effects of different phytochemicals on AFB1 in various experimental models (Table 2 and Table 3).

## 4. Berberine

Berberine is a plant alkaloid that has been utilized for generations in traditional Chinese and Indian medicine [91] and is known to protect against diabetes-related complications by enhancing insulin sensitivity, including mRNA expression in the human liver and skeletal muscle cell culture [92]. Considering that free radicals are formed as a result of lipid peroxidation in cell membranes [93], substances with antioxidant properties can remove free radicals because there are hepatoprotective [94]. According to Malekinezhad et al. this phytochemical (berberine) can reduce oxidative stress and inflammation in organs such as the liver, adipose tissue, kidney, and pancreas by increasing antioxidant enzymes and reducing pathological liver lesions [83]. Importantly, the anticoccidial activity of this compound has been shown against *Eimeria*-infected chickens [95] with some evidence indicating that it can improve gut wall morphology and microbiota in broiler chickens [96]. In situ and in vitro studies show that berberine can block broiler p-glycoprotein expression and its function [97].

## 5. Carvacrol and Trans-Cinnamaldehyde

Carvacrol is a key component of oregano oil (*Origanum glandulosum*), whereas trans-cinnamaldehyde is found in cinnamon bark (*Cinnamomum zeylandicum)*. Based on excellent pharmacological functions and availability, carvacrol and trans-cinnamaldehyde have been compounds of interest in poultry research [98]. Carvacrol and trans-cinnamaldehyde exhibit enhanced inhibitory properties against aflatoxin, as it downregulates the expression of genes (*Aflc*, *norA, nor1*, and *Ver1*) in the aflatoxin biosynthetic pathway for *A. flavus* and *A. parasiticus* [99]. This was supported by an in vivo study, where broiler chickens fed an aflatoxin-contaminated diet had improved histological lesions and antioxidant enzymes [100]. A follow-up study on this work showed that carvacrol and trans-cinnamaldehyde similarly protect chicken embryos against aflatoxin-induced toxicity [88]. The authors suggested that the potential diverse therapeutic mechanisms could be involved in the anti-aflatoxin effect of carvacrol, including CYP450 inhibition, antioxidant function, and modulation of aflatoxin detoxification enzymes.

## 6. Curcumin

Curcumin or diferuloylmethane is a polyphenolic compound isolated from *Curcuma longa* (turmeric) and widely used as a colourant and spice [101]. Reviewed information already shows that curcumin can improve biomarkers of oxidative stress in pathological conditions [102]. Curcumin research has grown in popularity among scientists due to its multiple beneficial pharmacological properties in rats, which include antioxidant [103,104], anti-inflammatory [105], anti-apoptosis [106], antitumor [107], and anti-cancer effects [108]. Curcumin protected mice from irinotecan-induced intestinal injury via modulating the oxidant/antioxidant balance [109] and has also been reported as a compound that alleviates AFB1 toxicity in the liver. Curcumin reduced ROS production and activated the nuclear factor erythroid 2-related factor 2/heme oxygenase-1 pathway in AFB1-induced chicken liver [56]. Furthermore, antioxidant gene upregulation increased [87], as curcumin consumption reduced the levels of liver enzymes, such as ALT and AST, in broilers [59]. Interestingly, restoration of histopathological liver lesions following damage by AFB1, as SOD and Nrf2 protein levels increase, has been reported in broilers and ducks [86,110,111]. Curcumin bioactivation of AFB1 reduced recombinant CYP2A6 in Arbor Acres broilers in a dose-dependent manner [112]. In broiler chickens, 444 mg/kg of curcumin in the diet lowered AFB1, resulting in the upregulation of hepatic genes involved in energy production and fatty acid metabolism, detoxification, coagulation, and immunological modulation [113]. It is well known that AFB1 increases the liver Phase-I enzyme CYP450 [38], but Muhammad et al. found that curcumin has the potential for inhibiting the CYP450 enzyme in chickens [86].

## 7. Proanthocyanidin and Thymoquinone

Proanthocyanidins are polyphenolic flavonoids primarily derived from grape seed extract. According to published works, these phytochemicals are attractive to many researchers [114,115]. These compounds can improve liver function in rats by strengthening the antioxidant system [116,117]. In another work, the immunological response in the mouse spleen induced by AFB1 was reduced [118]. The protective benefit of proanthocyanidins against AFB1-induced oxidative damage is achieved through modulating the Nrf2 signaling pathway in broiler chicken liver [85]. Furthermore, precise molecular mechanisms were reported in broilers by decreasing the inflammatory response and inhibiting nuclear factor-kappa B (NF-kB) expression. Deng et al. [119] reported that proanthocyanidin enhances GST enzyme activity and gene expression in cell culture, which is crucial for the detoxification of AFB1 and its toxic metabolites. Thymoquinone is a chemical compound found in *Nigella Sativa* seeds (also called Kalonji) that has antioxidant, anti-inflammatory, and hepatoprotective properties. Thymoquinone substantially restores the hepatic supply of GSH caused by AFB1 in mice [120] and appears to lower the hepatic level of MDA as a lipid peroxidation index, while GSH content is significantly restored for toxicity avoidance. Thymoquinone also prevented the broilers’ histological alterations caused by AFB1 [84].

## 8. Quercetin

Quercetin is an antioxidant that can be found in a wide range of fruits and vegetables such as blueberries, onions, curly kale, broccoli, and leeks [79,121]. By normalizing the redox state balance, quercetin consumption on AFB1-induced liver damage protects the hepatic tissues [82]. From the chemical view, only nonpolar phenolic compounds with no free hydroxyl groups are effective at detoxifying AFB1-mediated liver toxicity, compared to polyphenols with many hydroxyl groups [122].

Quercetin is known to inhibit free radicals, which contribute to protein and DNA oxidation as well as lipid peroxidation of biological membranes. Its high antioxidant power has been recognized as a treatment that significantly reduced AFB1-induced oxidative stress, as evidenced by a significant reduction in MDA, a marker of lipid peroxidation, and an increase in GSH, SOD, and CAT levels in the brain tissue of mice exposed to AFB1 and treated with quercetin [79].

## 9. Resveratrol

Resveratrol (trans-3,4,5-trihydroxystilbene) is a polyphenol that is a constituent of grapes, blueberries, and mulberries. The antioxidant potential of resveratrol minimizes the cytotoxic effects of oxidative stress in cells [123]. The harmful effects of free radicals and the contemporary ROS *production in chickens* have been reported to be negated by resveratrol [124]. In the in vivo and in vitro studies in mice, as well as in retrospective clinical trials, resveratrol’s biological effects have been intensively explored. Resveratrol lowers the frequency of diabetes and is an anti-cancer and anti-inflammatory agent with beneficial characteristics that are observed in mice and rat studies [125,126,127]. In summary, four studies that reported the ameliorating effect of resveratrol suggested a significant effect associated with aflatoxin exposure. Resveratrol ameliorated the AFB1 by increasing the antioxidative capacity of the enzymes that were downregulated by exposure to AFB1 [80]. Liu and Wang suggest that the kelch-like ECH-associated protein 1 (Keap 1)- and Nrf2-antioxidant response elements’ signaling pathways are both regulated by resveratrol [77]. Furthermore, resveratrol regulates gene expression to exert its biological function and has been reported to be an epigenetic factor [128], and a study in mice has indicated resveratrol’s regulatory role in DNA methylation [129], and histone methylation and acetylation [130].

## 10. Silymarin

The hepatoprotective effect of silymarin through various experimental models, including chickens, has been studied in acute liver intoxication induced by toxic agents [44,131,132]. Silymarin has hepatoprotective properties in the experimental intoxication with *Amanita phalloides* of humans [133], rats [134], and chickens [135] for competitive inhibition of α-amanitin uptake, in such a way as by inhibiting the murine hepatic cytochrome P450 detoxification system in the phase-I metabolism [136]. According to studies conducted in mice using Amanita phalloides, silybin inhibits the phase-I metabolism of the liver’s cytochrome P450 detoxifying system [137] and Rastogi et al. [138] reported that silymarin reversed changes in liver and serum in AFB1-intoxicated rats, indicating that it has a hepatoprotective action in preventing AFB1-induced injury. It is, however, reported that ALT activity was increased in the broilers treated with silymarin after there were exposed to AFB1 toxicity [136].

## 11. Summary and Future Perspectives

The harmful effects of aflatoxins, which are known to cause hepatotoxicity in cellular organisms, are well documented. Furthermore, evidence suggests that aflatoxins cause oxidative stress in broiler chickens, leading to cellular damage. As a result, an attempt to use bioactive substances to combat aflatoxins in the system, by providing a protective line of defense through antioxidant activity, has been made. As a result, studies must be conducted to determine whether aflatoxins can be controlled reliably or if phytochemicals must be introduced into the diet to reduce their toxicity. Already, various phytochemicals, including berberine, carvacrol, curcumin, proanthocyanidins, quercetin, resveratrol, and silymarin, have been studied for their potential therapeutic effects in ameliorating aflatoxin-induced cytotoxicity, by limiting oxidative stress via enhancing intracellular antioxidants. However, the available evidence is still limited, and additional studies are still required to protect against aflatoxin-induced cytotoxicity, especially in chickens.

## Figures and Tables

**Table 1 molecules-28-05369-t001:** Summary of the literature reporting on the impact of the oxidative stress caused by aflatoxins in broiler chickens.

Reference	Experimental Model	Dosage and Treatment Duration	Findings
[76]	Broilers (aged 35 d)	Received 100 µg/kg for 4 weeks	Aflatoxin B1 (AFB1) significantly decreased the relative body weights of *bursa of Fabricius*, antioxidant enzyme activities of total superoxide dismutase (SOD), catalase (CAT), glutathione peroxidase (GPx), glutathione transferase (GST), and total antioxidation capacity, while it increased the malonaldehyde (MDA) content.
[63]	Avian broilers (aged 120 d)	Received 100 µg/kg for 4 weeks	Antioxidant capacity (CAT, GPx, and glutathione (GSH) were reduced, and lipid peroxidation MDA and DNA damage (8-OHdG) were increased.Administration of AFB1-induced liver injury and decreased total protein and albumin concentrations. Induced hepatotoxicity by increasing alanine aminotransferase and aspartate aminotransferase activities. Also, mRNA and activity of enzymes responsible for the bioactivation of AFB1 into AFBO, which included cytochrome P450 (CYP450) A1, 1A2, 2A6, and 3A4, were negatively affected in liver microsomes after 2-week exposure to AFB1.
[77]	Culture media (Primary broiler hepatocytes)	Received 0.5, 1, 2.5, and 5 µmol/L	AFB1 evoked mitochondrial generation of reactive oxygen species (ROS). AFB1 also increased the percentage of apoptotic cells and the expression of caspase-9 and caspase-3. This was also consistent with the impairment of mitochondrial functions, activated ROS, induced apoptosis, and upregulated messenger RNA (mRNA) expression of nuclear factor erythroid 2-related factor 2 (Nrf2). Whereas, the mRNA expressions of nicotinamide adenine dinucleotide phosphate (NADPH), quinone oxidoreductase 1, SOD, and heme oxygenase 1 were reduced.

**Table 2 molecules-28-05369-t002:** A summary of in vivo studies on the role of phytochemicals in the alleviation of aflatoxin oxidative damage.

Reference	Phytochemical	Experimental Model Used (Animal, Strain)	Effective Dose and Treatment Duration	Main Findings
[78]	Resveratrol	Male albino rat’s Hepatocellular carcinoma (HCC) rat cells	Received 1 mg/kg resveratrol for 14 d	Resveratrol restored the levels of catalase (CAT) and glutathione peroxidase in aflatoxin B1 (AFB1)-induced HCC. Resveratrol adversely regulated the expression of nuclear factor kappa-light-chain-enhancer of activated B cells (NFκB) in AFB1-induced cells, which in turn increased the activity of the antioxidant enzymes.
[79]	Quercetin	Balb/c mice	Received 30 mg/kg quercetin for 35 d	Quercetin exerted a preventive role against oxidative stress by promoting antioxidative defence systems and limiting lipid peroxidation.
[80]	Resveratrol	Male rats- C57BL/6J	Received 500 mg/kg resveratrol for 35 d	Adverse hepatic function caused by AFB1 was ameliorated by resveratrol, via increasing hepatic antioxidative capacity and inhibiting the expression of a cleaved caspase-3 protein.
[81]	Curcumin	Rats	Received 15 mg/kg curcumin for 35 d	Curcumin ameliorated AFB1 via upregulation of antioxidant enzyme gene expression; thus, glutathione (GSH) availability was increased. Lipid peroxidation was normalised by a decrease in thiobarbituric acid reactive substances (TBARS) in the curcumin-treated group.
[82]	Quercetin	Sprague-Dawley rats.	Received 50 mg/kg quercetin for 35 d	Quercetin normalized the biochemical parameters such as glutathione peroxidase (GPx) and superoxide dismutase (SOD), as well as fatty acid synthase and tumor necrosis factor-alpha (Tnf-α) gene expression, in the liver tissue. It regulated the alteration of gene expression and improved the histopathological and histochemical picture of the liver.

**Table 3 molecules-28-05369-t003:** A summary of studies reporting the effect of phytochemicals in ameliorating oxidative damage induced by aflatoxins in broilers.

Reference	Phytochemical	Experimental Model Used (Animal)	Effective Dose and Intoxication Method	Main Findings
[83]	Berberine	1-d Ross 308	Received 200, 400, and 600 mg/kg berberine for 42 d	Supplementation partially/completely reversed alanine aminotransferase (ALT), aspartate transferase (AST), and gamma-glutamyl transferase (GGT) levels in blood serum. The dose of 600 mg/kg of the compound even showed an enhanced effect in improving liver function, and antioxidant status after exposure to aflatoxin B1 (AFB1).
[59]	Curcumin	Arbor Acres	Received 300 mg/kg curcumin diet for 28 d	Decreased liver enzymes ALT, AST, and Y-glutamyltransferase increased by AFB1. Furthermore, it ameliorated histopathological liver lesions.
[84]	Thymoquinone	1-d Ross 308	Received 300 mg/kg thymoquinone diet for 28 d	Thymoquinone ameliorated aflatoxicosis lesions and deteriorations in biochemistry levels. It alleviated liver injury by inhibiting or reducing the bioactivation of AFB1 through phase-I nuclear receptors and cytochrome P450 enzyme modulation.
[85]	Proanthocyanidin	1-d Cobb	Received 250 mg/kg proanthocyanidin diet for 35 d	Proanthocyanidin enhances the antioxidant enzymes by decreasing the malonaldehyde (MDA) content. Also, it enhanced the expression of antioxidant genes. PC ameliorated AFB1 and induced oxidative stress by modulating the antioxidant defense.
[86]	Curcumin	1-d Arbor Acres	Received 450 mg curcumin for 28 d	Partial improvement in liver histology and the MDA content was decreased with an increase in superoxide dismutase (SOD) and nuclear factor erythroid 2-related factor 2 (Nrf2) protein levels.
[87]	Curcumin	1-d Arbor Acres	Received 150, 300, and 450 mg/kg curcumin diet for 28 d	Up-regulated Nrf2 and downstream genes’ messenger RNA (mRNA) expression levels. Induced liver injury via enhancing phase-II enzyme expression and activity. However, it increased AFB1–GSH conjugation in vitro in liver cytosol.
[88]	Carvacrol and trans-cinnamaldehyde	1-d white Leghorn eggs	Received 0.1 % carvacrol and trans-cinnamaldehyde injection for 18 d	Supplementation decreased the aflatoxin embryotoxicity by improving the embryo size.
[89]	Resveratrol	Broilers	Received 0.5% and 1% resveratrol diet for 42 d	Increased the activity of the oxidative stress enzymes and improved the total antioxidant capacity and protein.
[90]	Silymarin phytosome	1-d broiler	Received 600 mg/kg silymarin diet for 35 d	Increased ALT activity and decreased histopathological changes in the liver.

## Data Availability

Not applicable.

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
