# Peer review of "Pathological Role of Oxidative Stress in Aflatoxin-Induced Toxicity in Different Experimental Models and Protective Effect of Phytochemicals: A Review"

_molecules, 2023, doi:10.3390/molecules28145369_

Round 1
Reviewer 1 Report
GENERAL COMMENTS
The authors have summarized how some phytochemicals (11) can be used to protect against aflatoxin-induced toxicity in broiler chicken. The paper per se is interesting since Aflatoxin B1 (AFB1) is distributed worldwide, it belongs to group 1 carcinogen and can cause hepatotoxic, teratogenic, and immunosuppressive effects, so it is significant to reduce human exposure to it. As authors stated, phytochemicals can modulate gene expression, enzymatic activity and protect against oxidative stress. The review focused on Berberine, Carvacrol and trans-cinnamaldehyde, Carnosic acid, Curcumin, Piperine, Proanthocyanidin and Thymoquinone, Quercetin, Resveratrol and Sylmarin. The authors thus conclude that more studies should be carried out to confirm the modulate effect of phytochemicals.
SPECIFIC COMMENTS FOR REVISION
-I have found some typographical errors in the document. I point out the following, but there are likely others, so a closer read of the manuscript is warranted:
- Check spaces throughout the manuscript (e.g., line 21, 40, 127, 130, 176, 177, 199, 210, 260, 270, table 2…)
- Check punctuation points (e.g., line 143, heading table 2, in table 2 in main findings of Resveratrol, footnote figure 8…)
- Check abbreviations. The full name should be described before abbreviation (e.g., GST in line 136, GGT, ALT and AST in table 3…) and once an abbreviation is named, it should be continued throughout the manuscript (e.g., BBR in line 158 then is not used at line 163, aflatoxin B1 in line 269…). Authors didn’t indicate the abbreviation of Carnosic acid.
- Scientific names of plants should be in italics (e.g., Amanita phalloides in line 284…)
- Check references (there is an error in number 80, in some of them the journal is not indicated e.g., 35, 84…).
- The quality of the images should be improved and keep the aspect ratio.
why do the authors show a figure of all compounds except carvacrol and trans-cinnamaldehyde?
- The reason why the authors selected these phytochemicals should be clarified.
- Table 1: Please clarify dosage (e.g., reference 65, dosage of 100 µg/kg of which compound? Reference 67, dosage of 25, 50, 75 and 100%, what does that mean?).
- Table 2 and 3: If the studies were performed in presence of AFB1, the concentration of the mycotoxin should be also indicated at the table.
- The division of tables 2 and 3 (alleviation and ameliorating) seem confusing in my opinion. I would recommend change it, for example by assay (in vitro/in vivo), by phytochemical...
- The information related to Piperine is scarce and not related to AFB1. The authors do not name this compound in the tables provided. I would recommend removing this compound or adding more information. Similar situation with Carnosic acid, the discussion is better explained but little information is given, and it is not even named at the tables.
- Line 166. “Malekinezhad et al. [72] reported that broilers fed an aflatoxin-contaminated feed had or showed increased antioxidant enzymes and histopathology.” Please, clarify histopathology in this context. How did it change?
- Most of the references are from more than 5 years ago, if there are no current studies combining ABF1 with phytochemicals, it would be interesting to indicate it as a conclusion and to elucidate what the reason may be.
Author Response
Manuscript ID: molecules-2409010
Reviewer 1
The authors have summarized how some phytochemicals (11) can be used to protect against aflatoxin-induced toxicity in broiler chicken. The paper per se is interesting since Aflatoxin B1 (AFB1) is distributed worldwide, it belongs to group 1 carcinogen and can cause hepatotoxic, teratogenic, and immunosuppressive effects, so it is significant to reduce human exposure to it. As authors stated, phytochemicals can modulate gene expression, enzymatic activity and protect against oxidative stress. The review focused on Berberine, Carvacrol and trans-cinnamaldehyde, Carnosic acid, Curcumin, Piperine, Proanthocyanidin and Thymoquinone, Quercetin, Resveratrol and Sylmarin. The authors thus conclude that more studies should be carried out to confirm the modulate effect of phytochemicals.
We appreciate the positive response and the time the reviewer took to assess the manuscript. Please note, revision of tables, enhanced formatting, updating references, as well as addressing therapeutic doses of the compound being used, has been provided. Please note, all implemented changes have been using track changes, as well as highlighted in yellow within the manuscript. We hope you find the revised version of our manuscript acceptable in its current form.
Specific comments for revision
I have found some typographical errors in the document. I point out the following, but there are likely others, so a closer read of the manuscript is warranted:
Check spaces throughout the manuscript (e.g., line 21, 40, 127, 130, 176, 177, 199, 210, 260, 270, table 2…)
Thank you for this important comment, please note, the formating of the manuscript, including typographical errors and spaces have been corrected throughout the manuscript.
Check punctuation points (e.g., line 143, heading table 2, in table 2 in main findings of Resveratrol, footnote figure 8…)
Thank you, and consistent with the above comment, we have corrected the punctuation points in line 143, table 2 heading and throughout the manuscript.
Check abbreviations. The full name should be described before abbreviation (e.g., GST in line 136, GGT, ALT and AST in table 3…) and once an abbreviation is named, it should be continued throughout the manuscript (e.g., BBR in line 158 then is not used at line 163, aflatoxin B1 in line 269…). Authors didn’t indicate the abbreviation of Carnosic acid.
Thank you for this important point, as we have fully described the abreviaions, including GST, GGT, ALT, and AST. In addition, BBR has been used after the abbreviation and Aflatoxin B1 changed to AFB1, while carnosic acid has also been removed because there was no study on broiler chickens
Scientific names of plants should be in italics (e.g., Amanita phalloides in line 284…)
This is important comment that appreciate, and we have italicized Amanita phalloides and other scientific names throughout the manuscript.
Check references (there is an error in number 80, in some of them the journal is not indicated e.g., 35, 84…).
Thank you, all references have been fixed.
The quality of the images should be improved and keep the aspect ratio. Why do the authors show a figure of all compounds except carvacrol and trans-cinnamaldehyde?
We appreciate this comment from the reviewer. Please note, all figures/images for chemical compounds have been removed since we believe they do not add value to the mansucript, which is also consistent with addressing the commenf from the other reviewer.
The reason why the authors selected these phytochemicals should be clarified.
Thank you for this important comment and we agree with the reviewer, as such we have provided a statement for selecting/focusing on the phytochemicals covered within the manuscript.
Table 1: Please clarify dosage (e.g., reference 65, dosage of 100 µg/kg of which compound? Reference 67, dosage of 25, 50, 75 and 100%, what does that mean?).
This is a very impotant comment, thank you. Please note, we have clarrified and given more details on the dosages used for all discussed/cited literature within Table1.
Table 2 and 3: If the studies were performed in presence of AFB1, the concentration of the mycotoxin should be also indicated at the table.
This is a very important comment, thank you. As such, we have revised the Tables to provide the concentration of AFB1 or mycotoxin.
The division of tables 2 and 3 (alleviation and ameliorating) seem confusing in my opinion. I would recommend change it, for example by assay (in vitro/in vivo), by phytochemical...
We appreciate this important comment from the reviewer. As such, we have divided/seperated the tables based on in vitro and in vivo evidence.
The information related to Piperine is scarce and not related to AFB1. The authors do not name this compound in the tables provided. I would recommend removing this compound or adding more information. Similar situation with Carnosic acid, the discussion is better explained but little information is given, and it is not even named at the tables.
We appreciate this important comment and agree with the reviewer. As such, we have removed information of piperine, while content was added for carnosic acid, including amendments within the table/s.
Line 166. “Malekinezhad et al. [72] reported that broilers fed an aflatoxin-contaminated feed had or showed increased antioxidant enzymes and histopathology.” Please, clarify histopathology in this context. How did it change?
Thank you, we have revised text to be more specific where necessary, including fully describing the histopathological changes caused by exposure to AFB1.
Most of the references are from more than 5 years ago, if there are no current studies combining ABF1 with phytochemicals, it would be interesting to indicate it as a conclusion and to elucidate what the reason may be.
We agree with the reviewer, and as such we have revised the text, and updated the citations, including the reference list, especially focusing on recent information, published within the past 5 years.
Reviewer 2 Report
You can find review in Word file.
Suggestions are made in order to make manuscript acceptable. There are two parts, one is about mechanism of intoxication, and second is phytotherapy. I have suggested title of the manuscript that is uniting both of this parts. Please, for each effect written you have to mention on what kind of experimental model it is obtained (humans, mice, chickens...).

English language quality is not acceptable in present form.
Author Response
Manuscript ID: molecules-2409010
Reviewer 2
Suggestions are made in order to make manuscript acceptable. There are two parts, one is about mechanism of intoxication, and second is phytotherapy.
We appreciate the positive response and the time the reviewer took to assess the manuscript. Please note, revision of tables, amendment of title, enhanced formatting, updating references, as well as addressing therapeutic doses of the compound being used, has been provided. Please note, all implemented changes have been using track changes, as well as highlighted in yellow within the manuscript. We hope you find the revised version of our manuscript acceptable in its current form.
I have suggested title of the manuscript that is uniting both of this parts. Please, for each effect written you have to mention on what kind of experimental model it is obtained (humans, mice, chickens...).
Thank you for this important comment, please note the title of the manuscript has been revised as suggested “Pathological role of oxidative stress in aflatoxin-induced toxicity in different experimental models and protective effect of phytochemicals: A review”
Commented [A3]Start your paper with this sentence starting in line 47 and the next one starting in line 49. The continue as you started.
We appreciate this important comment, and as suggested by the reviewer, we have started the paper with line 47 as.
Commented [A4]: Please use another adjective like harmful…
Thank you, and as suggested, we have used the adjective, “Harmful”.
Commented [A5]: Put this in paragraph above.
Thank you for this important comment, as suggested, the highlighted sentence was moved to the paragraph mentioned above.
Commented [A6]: By looking in cited literature, I could not find that AFB1 is classified in Group 1.
Thank you for this important comment, we have revised the mansucript to indicate that AFB1 is classified in Group 1. This has been added as part of the introduction.
Commented [A7]: Start new paragraph from this point
Thank you, we have noted, and addressed this comment.
Commented [A8]: This is not understandable, please check this and make correction.
Thank you, we have noted, and addressed this comment.
Commented [A9]: In previous sentence you were mentioning scientifical methods. I gues this is a method used in everyday life of each person, so please make difference in expression.
Thank you, we have noted, and addressed this comment.
Commented [A10]: This reference does not support written statement, in page 7 of [26] it is written “Dietary inclusion of curcumin (0.6-0.9 g/kg diet), improved the adverse effect of aflatoxins in blood glucose values of broiler chickens”. Please delete this reference and find another one that supports your statement.
Thank you for noting this error, the reference was deleted and revised as suggested.
Commented [A11]: I could not find support for what is written in this sentence in references [30,31], please cite other one that support
Thank you for this comment, and as suggested, we have removed citations [30, 31] and new citations added, as suggested by the reviewer.
Commented [A12]: Please create section Material and Methods and place this in it.
We appreciated this comment, and as sugested, we have created a methodology section.
Commented [A13]: Please give reference for this statement.
Thank you for this important comment, we have added a reference to the suggested statement.
Commented [A14]: Broiler chickens are not even mentioned in cited reference. Please give another reference for statement in this sentence.
Thank you for this important comment, and as suggested we have revised the information, especially adding the precise reference.
Commented [A15]: In this experimental work rats were used as test animals, not chickens.
Thank you for this important comment, and cited information has been corrected.
Commented [A16]: In this experimental work rats were used as test animals, not chickens.
Thank you for this important comment, and cited information has been corrected.
Commented [A17]: This experimental work was conducted on ducks.
Thank you for this important comment, and cited information has been corrected.
Commented [A18]: Results of this work are based on the Ghanaians.
Thank you for this important comment, and cited information has been corrected.
Commented [A19]: In this experiment turkeys were used as test animals.
Thank you for this important comment, and cited information has been corrected.
Commented [A20]: Experimental results in this work were obtained on humans.
Thank you for this important comment, and cited information has been corrected.
Commented [A21]: [48] is on chickens
Thank you for this important comment, and cited information has been corrected.
Commented [A22]: Murine livers and bovine peripherial blood are not mentioned in cited references 50-52
Thank you for this important comment, and cited information has been corrected.
Commented [A23]: In this reference aflatoxin is not even mentioned, they were writing about deoxynivalenol. Please delete this reference.
Thank you for this important comment, and cited information has been deleted.
Commented [A24]: This work is about glutathione conjugates with mycotoxin 4-deoxynivalenol. Please delete this reference and replace them with ones that are about aflatoxins.
Thank you for this important comment, and cited information has been corrected.
Commented [A25]: Aflatoxins are not even mentioned in this reference. Please delete it.
Thank you for this important comment, and cited information has been deleted.
Commented [A26]: This experiment was conducted on rats.
Thank you for this important comment, and cited information has been corrected.
Commented [A27]: This reference does not support statement in sentence that you wrote, it written “Turkeys exhibit greater sensitivity to this toxin when compared to other poultry species, particularly chickens”, not that they have “excellent phase I liver enzymes”.
Thank you for this important comment, and cited information has been corrected.
Commented [A28]: In reference [61] mammals are not even mentioned, it is about humans. You can not presume that it is for all mammals. Please delete this sentence or find another reference that supports what is written.
Thank you for this important comment, and the sentence was deleted.
Commented [A29]: There is not mentioning abour Nrf2 in reference 62.
Thank you for this important comment, and cited information has been corrected.
Commented [A30]:
Thank you for this important comment, and cited information has been corrected.
Commented [A31]: It is said in Table caption that it is about aflatoxin
Thank you for this important comment, and cited information has been corrected.
Commented [A32]: Culture medium was used
Thank you for this important comment, and cited information has been corrected.
Commented [A33]: In this experimental model was used mixture of different toxins, if all of them does not belong to the group of aflatoxins than you should either delete this row or change capitation of table aflatoxins into mycotoxins.
Thank you for this important comment, and the row was deleted.
Commented [A34]: It would be helpful to add for each
Thank you for this important comment, and the cited information has been amended to be more clear to the reader.
Commented [A35]: There is no data about duration in this column, however you can delete “and duration”
Thank you for this important comment, and cited information has been corrected.
Commented [A36]: There is very clear sentence in this reference that is explaining mechanism of activation and detoxification “It is well understood that liver cytochrome p450 enzymes are responsible for AFB1 bioactivation, while phase-II enzymes regulated by the transcription factor nuclear factor-erythroid-2-related factor 2 (Nrf2) are involved in detoxification of AFB1”
Thank you for this important comment, suggested information has been added.
Commented [A37]: It is not understandable what you wanted to say, please write again this sentence to be clear.
Thank you for this important comment, and cited information has been corrected to be made clear to the reader.
Commented [A38]: There is no need for chemical formulas of phytochemicals
We agree with the reviewer, all information/formulas of phytochemicals has been removed.
Commented [A39]: CR and TC are not even mentioned in this work, please delete this reference.
Thank you for this important comment, and cited information has been deleted.
Commented [A40]: CR and TC are not even mentioned in this work, please delete this reference.
Thank you for this important comment, and cited information has been deleted.
Please support statement in this sentence by adequate reference/s.
Thank you for this important comment. Cited information has been amended by providing more precise citations.
Commented [A41]: Please give reference.
Thank you for this important comment, and cited information has been corrected.
Commented [A42]: This reference is not even mentioning free radical scavenger ability of carnosi acid.
Thank you for this important comment, and cited information has been corrected.
Commented [A43]: There is no need for chemical formula and it should be deleted.
Please note, consistent with the above comment, the chemical formular has been deleted.
Commented [A44]: Please written this sentence again but this time more precisely and clearly to have this form: “… antioxidant in (humans, rats, …) [reference], anti-inflammatory in (human, rats, chickens…) [reference], …
Thank you for this important comment, the aforementioned information has been corrected, as suggested by the reviewer.
Commented [A45]: Please rearrange this sentence to be more clear.
Thank you for this important comment, and the cited information has been deleted.
Commented [A46]: This sentence is not understandable, please use another appropriate and more descriptive verb.
Thank you for this important comment, and the cited information has been amended to be more clear to the reader.
Commented [A47]: They did not wrote that curcumin has carcinogenic and hepatotoxic potential, please read again more carefully and write this sentence again.
Thank you for this important comment, and the information has been amended to be more clear to the reader.
Commented [A48]: There is no need for this figure, please delete.
Thank you for this important comment, and the figure has been removed.
Commented [A49]: There is no need for chemical formula, please delete.
Thank you, and consistent witht the above comment, all chemical formulas have been removed.
Commented [A50]: Please give reference for statement in this sentence.
Thank you for this important comment, and the cited information has been added as recommended.
Commented [A51]: There is no need for chemical formulas, please delete.
Thank you, and consistent witht the above comment, all chemical formulas have been removed.
Commented [A52]: There is no need for chemical formulas, please delete.
Thank you, and consistent witht the above comment, all chemical formulas have been removed.
Commented [A53]: Blueberries, curly kale and leeks are not mentioned in cited work, please delete or add another reference that supports statement in this sentence.
Thank you, and consistent witht the above comment, information has been revised and precise citation provided.
Commented [A54]: Start new paragraph from here.
Thank you for this important paragraph, and the new paragraph has been removed.
Commented [A55]: There is no need for chemical formula, please delete.
Thank you, and consistent witht the above comment, all chemical formulas have been removed.
Commented [A56]: Please add for each effect what animals were used in experiment.
Thank you for this imporant comment, and suggested information hase been added.
Commented [A57]: Blueberries and mulberries are not mentioned in cited work, please add another reference that is mentioning presence of resveratrol in these berries.
Thank you, and consistent witht the above comment, information has been revised and precise citation provided.
Commented [A58]: In work 126 quails were experimental animals, please animals that are used in each research that you are mentioning.
Thank you, and consistent witht the above comment, information has been revised and precise citation provided.
Commented [A59]: Please reformulate this sentence to be more clear.
Thank you, we have revised information to be more precise and clear to the reader.
Commented [A60]: What current review, the cited one? If you meant on this sentence could be formulated like: “The review paper of Liu&Wang suggests …”
Thank you, and consistent witht the above comment, information has been revised and precise citation provided.
Commented [A61]: In this experiment was performed on mice.
Thank you, and consistent witht the above comment, information has been revised and precise citation provided.
Commented [A62]: This is what it is written in Abstract of cited work “RSV could reduce the content of MDA and elevate the T-SOD activity in serum”.
Thank you, and consistent witht the above comment, information has been revised and precise citation provided.
Commented [A63]: There is no need for chemical formula, please delete.
Thank you for this important comment, the chemical formula has been removed.
Commented [A64]: These are also presumed mechanisms of action mentioned in same work [79]: “Some authors reported that silymarin has hepatoprotective properties in experimental intoxication with Amanita phalloides, for a competitive inhibition of α-amanitin uptake (Kro¨ncke et al., 1986). It is reported that silybin inhibits the murine hepatic cytochrome P450 detoxification system in the phase I metabolism (Baer-Dubowska et al., 1998). Rastogi et al. (2000) reported that silymarin reversed changes in liver and serum in AFB1 intoxicated rats, indicating that it has a hepatoprotective action in preventing AFB1 induced injury.”
Thank you, information has been revised and precise citation provided.
Commented [A65]: This sentence is not understandable, please rearrange it to be clear.
Thank you, and consistent witht the above comment, information has been revised and precise citation provided.
Commented [A66]: Please express yourself clear.
Thank you, and consistent witht the above comment, information has been deleted after revising
Commented [A67]: There is no need for chemical formula, please delete.
Thank you, the chemical formula has been removed.
Commented [A68]: Please write this section again. While you are writing it please have on mind all that you wrote above and write it as concisely and clearly as possible.
Thank you, and consistent witht the above comment, information has been revised and precise citation provided.
Commented [A69]: Please check all references once again and write them accordingly to the given instructions for authors. i.e. Author 1, A.B.; Author 2, C.D. Title of the article. Abbreviated Journal Name Year, Volume, page range
Thank you for this important comment, and guidelines have been followed for revised and precise citations.
Round 2
Reviewer 1 Report
The authors have responded and improved the manuscript following the recommendations given.